# EPS364, a Novel Deep-Sea Bacterial Exopolysaccharide, Inhibits Liver Cancer Cell Growth and Adhesion

**DOI:** 10.3390/md19030171

**Published:** 2021-03-22

**Authors:** Yun Wang, Ge Liu, Rui Liu, Maosheng Wei, Jinxiang Zhang, Chaomin Sun

**Affiliations:** 1CAS and Shandong Province Key Laboratory of Experimental Marine Biology & Center of Deep Sea Research, Institute of Oceanology, Chinese Academy of Sciences, Qingdao 266071, China; wangyunxixi@yeah.net (Y.W.); liuge@qdio.ac.cn (G.L.); liurui@qdio.ac.cn (R.L.); weimaosheng18@mails.ucas.ac.cn (M.W.); 2Laboratory for Marine Biology and Biotechnology, Qingdao National Laboratory for Marine Science and Technology, Qingdao 266071, China; 3College of Earth Science, University of Chinese Academy of Sciences, Beijing 100049, China; 4Center of Ocean Mega-Science, Chinese Academy of Sciences, Qingdao 266071, China; 5Qingdao Vland Biotech Inc., Qingdao, 266102, China; jinxiang.zhang@gmail.com

**Keywords:** exopolysaccharide, antitumor, FGF19-FGFR4 signaling, cell growth, cell adhesion

## Abstract

The prognosis of liver cancer was inferior among tumors. New medicine treatments are urgently needed. In this study, a novel exopolysaccharide EPS364 was purified from *Vibrio alginolyticus* 364, which was isolated from a deep-sea cold seep of the South China Sea. Further research showed that EPS364 consisted of mannose, glucosamine, gluconic acid, galactosamine and arabinose with a molar ratio of 5:9:3.4:0.5:0.8. The relative molecular weight of EPS364 was 14.8 kDa. Our results further revealed that EPS364 was a β-linked and phosphorylated polysaccharide. Notably, EPS364 exhibited a significant antitumor activity, with inducing apoptosis, dissipation of the mitochondrial membrane potential (MMP) and generation of reactive oxygen species (ROS) in Huh7.5 liver cancer cells. Proteomic and quantitative real-time PCR analyses indicated that EPS364 inhibited cancer cell growth and adhesion via targeting the FGF19-FGFR4 signaling pathway. These findings suggest that EPS364 is a promising antitumor agent for pharmacotherapy.

## 1. Introduction

Liver cancer is listed as the fourth leading cause of cancer death and the sixth most common neoplasm, with approximately 782,000 deaths and 841,000 new cases each year [1]. In the past decades, significant advances have been made in the treatment of liver cancer, including surgical treatments, locoregional treatments and systemic treatments. However, the major shortcomings of these treatments are progression after effective chemoembolization and recurrence after ablation or resection [2]. Compared with other tumors, the prognosis of liver cancer still remains inferior. Liver cancer is not only a highly chemotherapy-resistant tumor, but the applicability of most chemotherapy regimens is severely limited by the underlying liver disease [3]. Therefore, new treatments are urgently needed.

Fortunately, diverse and novel marine ecosystems provide the possibility to discover and develop new therapeutic agents with unique mechanisms [4]. In the past few decades, small-molecule compounds from marine organisms have been widely studied for drug use [2,3,4,5]. Besides, a lot of marine-derived anticancer drugs have successfully entered the market or have been approved for clinical trials, including glembatumumab vedotin, pinatuzumab vedotin, lurbectedin and eribulin [5]. Of note, macromolecular compounds especially microbial exopolysaccharides from marine organisms, possess a variety of underlying pharmaceutical activities, such as antitumor, anti-inflammatory and antiviral [6,7,8,9,10]. However, macromolecular compounds have not been thoroughly studied in drug research because of the complexity and difficulty of the purification of the structures [6,7,8,9].

Exopolysaccharides (EPS) are high molecular weight carbohydrate polymers synthesized via intracellular or extracellular pathways and then secreted extracellularly by microorganisms [9,10]. In our previous study, we found that a deep-sea seamount bacterial exopolysaccharide EPS11 inhibited cancer cell growth via blocking cell adhesion and stimulating anoikis [11,12]. After that, we isolated a large number of bacteria from different deep-sea environments, including cold seep, hydrothermal vents and the abyss. However, whether these bacteria could produce exopolysaccharides showing antitumor activity with a similar mechanism is not clear. In the present study, we purified a novel exopolysaccharide EPS364 from a deep-sea cold seep bacterium *Vibrio alginolyticus* 364, and the characterization and antitumor activity of EPS364 were further examined. Our aim was to uncover the specific and common antitumor mechanisms of exopolysaccharide EPS364. Notably, EPS364 significantly inhibited cancer cell growth and adhesion via targeting the FGF19-FGFR4 signaling pathway, indicating that EPS364 could be developed as an anti-liver cancer drug candidate. Moreover, EPS364 possessed antitumor activity via attenuating cancer cell adhesion, similar to that shown in EPS11 [11,12], indicating that similar deep-sea extreme conditions might facilitate bacteria to produce similar functional natural products.

## 2. Results

### 2.1. Isolation, Extraction and Purification of Exopolysaccharide EPS364

The crude polysaccharide was firstly purified through a DEAE Sepharose Fast Flow column, and the active crude EPS364 was obtained (Figure 1A). The active crude EPS364 was then loaded onto a Sephadex G-100 column, and the purified EPS364 was obtained (Figure 1B). The purified EPS364 powder was obtained with a yield of 20 mg/L (1-L fermentation broth yielded 20-mg EPS364). The UV spectrum of EPS364 was detected using a NanoPhotometer^®^ spectrophotometer at a range of 200–800 nm. As shown in the UV spectrum, the purified EPS364 had no obvious absorption at 260 or 280 nm (Figure 1C), indicating the absence of nucleic acids and proteins in EPS364. These data suggest that EPS364 is a pure polysaccharide.

### 2.2. Characterization of EPS364

#### 2.2.1. Molecular Weight (Mw) and Monosaccharide Composition of EPS364

The Mw and monosaccharide composition are important parameters that influence the bioactivity of a polysaccharide [13]. As shown in Figure 1D, EPS364 was eluted as a single symmetrical sharp peak with a retention time at 17.973 min, indicating that EPS364 was a homogeneous polysaccharide. According to the calibration curve of standards (Lg Mw = −0.2552, Rt = +8.7723, R^2^ = 0.9894; Mw represents the relative molecular weight, and Rt represents the retention time), the average Mw of EPS364 was estimated to be 14.8 kDa. According to comparing with the retention time of standard monosaccharides, we concluded that EPS364 consisted of mannose, glucosamine, gluconic acid, galactosamine and arabinose with a molar ratio of 5:9:3.4:0.5:0.8 (Figure 1E). 

#### 2.2.2. Fourier-Transform Infrared (FTIR) Spectrum Analysis

The FTIR spectrum analysis revealed distinctive chemical bands with typical absorption peaks [14]. Therefore, the FTIR spectrum analysis plays a vital role in the structural characterization analysis of polysaccharides. The FTIR spectrum of EPS364 is shown in Figure 2. The peaks at 3272 and 1043 cm^−1^ were attributed to the stretching vibrations of the O-H and C-O groups. The absorption peak at 2929 cm^−1^ was attributed to the stretching vibration of C-H [15]. The strong absorption peak near 1044 cm^−1^ was attributed to the C-O-C and C-O groups, which suggested the presence of a pyranose ring [14]. The absorption peaks at 1644 and 1403 cm^−1^ revealed the carboxylate groups of uronic acid residues [16]. The peak at 1312 cm^−1^ corresponded to C-H bending vibrations. In addition, the peak at around 1550 cm^−1^ corresponded to the carbonyl C=O stretching vibration and N-H bond [13,14]. Moreover, the absorption peak around 893 cm^−1^ corresponded to the β-linked glycosidic bond [13,16,17]. The absorption peak at 1246 cm^−1^ was caused by the P=O stretching vibration in the phosphate group [18]. These data suggest that EPS364 is a β-linked and phosphorylated polysaccharide.

#### 2.2.3. Scanning Electron Microscopy (SEM) and Energy-Dispersive Spectrum (EDS) Observations of EPS364

Biomacromolecule surface morphology as an important characteristic is associated with the other physical properties of a polymer, such as the rheological behavior and water holding capability [19]. The surface morphology of EPS364 was visualized using SEM. As shown in Figure 3A–C, EPS364 exhibited a regular porous network structure. The porous network structure increases the viscosity and water holding capacity of the polysaccharide, making EPS364 more suitable for pharmaceutical applications [20].

EDS detection revealed that EPS364 contained six major elements, except H (Figure 3D,E). The six elements were C, O, P, N, Na and Ca with the proportions of 48%:37%:4%:3%:6%:2%. C and O constitute long chains of the polysaccharide. P belongs to the phosphate group, according to the FTIR analysis. N belongs to glucosamine and galactosamine, according to the monosaccharide analysis. Na and Ca bind to the carboxyl, phosphate or hydroxyl groups of the polysaccharide [20].

All characterization analyses showed that EPS364 was a novel polysaccharide that possesses different structural characteristics from other polysaccharides. Besides, we found interesting features of EPS364, such as the relative lower Mw of 14.8 kDa, phosphorylation and porous network structure. These features increase the water solubility, viscosity and water holding capacity of the polysaccharide, indicating that EPS364 possesses excellent characteristics for future drug development.

### 2.3. Antitumor Activities of Exopolysaccharide EPS364

#### 2.3.1. EPS364 Inhibited Cell Proliferation and Adhesion

To investigate the therapeutic potential of EPS364, we detected its effects on human normal liver HL-7702 cells, liver cancer Bel-7402 cells and liver cancer Huh7.5 cells. As shown in Figure 4A, EPS364 preferentially killed cancer cells, including human liver cancer Bel-7402 cells and Huh7.5 cells, compared with human normal liver HL-7702 cells in a dose-dependent manner after treatment for 48 h. When the concentration was above 0.5 mg/mL, EPS364 better inhibited Huh7.5 cells. Therefore, we chose Huh7.5 cells as our model to investigate the molecular mechanism of EPS364. As shown in Figure 4B, the relative Huh7.5 cell viability decreased from 100% to 28.4% after EPS364 treatment (0.5 mg/mL) for different times (0–72 h). EPS364 inhibited the Huh7.5 cell growth in a time-dependent manner. According to Figure 4A,B, EPS364 exhibited significant antitumor activity in dose- and time-dependent manner. 

To investigate the action mode of EPS364, we observed the cell morphology changes in human normal liver HL-7702 cells and liver cancer Huh7.5 cells by inverted-phase contrast microscope and SEM after EPS364 treatment. To better observe the primal cell morphology changes, HL-7702 cells and Huh7.5 cells were treated with high concentrations of EPS364 (0–1 mg/mL) for 12 h, respectively. In Figure 4C(a,b), the obvious cell morphology change of HL-7702 was not observed. In the contrast, the Huh7.5 cell morphology was changed after EPS364 treatment. From the inverted-phase contrast microscope observation, Huh7.5 cells aggregated, attenuated the cell adhesion capability and detached from the extracellular matrix in a dose-dependent manner (Figure 4C(c)). From the SEM observation, Huh7.5 cells attenuated the filopodia structures and adhesion capability in a dose-dependent manner (Figure 4C(d)). When the concentration of EPS364 increased to 1 mg/mL, almost all the Huh7.5 cells lost their filopodia structures and cell adhesion capability after 12 h of treatment. The filopodia structures were considered to play vital roles in cell adhesion. The loss of the filopodia structures reduced the cell adhesion capability [21]. These results suggest that EPS364 inhibits cell proliferation and adhesion. 

#### 2.3.2. EPS364 Induced Apoptosis, Loss of Mitochondrial Membrane Potential (MMP) and Reactive Oxygen Species (ROS)

To investigate whether the growth inhibition effect was related to the induction of apoptosis, we performed the apoptosis analysis with high concentrations of EPS364 (0–2 mg/mL) for a short time (12 h). To observe the primal changes of the MMP and ROS, Huh7.5 cells were treated with high concentrations of EPS364 (0–2 mg/mL) for a short time (12 h). The results of the flow cytometry analysis (Figure 5A,B) showed that the apoptosis of Huh7.5 cells was remarkably induced after EPS364 treatment. Huh7.5 cells with EPS364 treatment resulted in increases of early apoptosis and late apoptosis, from 5.2% to 8.5% and 4.3% to 13.5%, respectively. These data indicated that cell apoptosis might contribute to the growth inhibition effect induced by EPS364.

The MMP as an evaluation indicator reflects the mitochondrial function [22]. The loss of the MMP is considered to be a marked event in the mitochondrial-mediated apoptosis pathway [23]. As shown in Figure 5C, EPS364 induced the dissipation of the MMP. These data indicated that the decrease of the MMP might participate in the apoptosis induced by EPS364.

High levels of ROS cause oxidative damage of DNA, proteins and lipids in tandem and, as a consequence, induce cancer cell apoptosis and death [24]. Since the mitochondria is one of the largest contributors to the endogenous ROS pool and the dissipation of the MMP was associated with the generation of ROS [25], we then examined the intracellular ROS. As shown in Figure 5D, EPS364 induced the production of ROS. These data indicated that the generation of ROS might also contribute to apoptosis induced by EPS364.

#### 2.3.3. Proteomic Analysis Growth Inhibition of Huh7.5 Cells after EPS364 Treatment

In order to investigate the antitumor mechanism of EPS364, we conducted a proteomic analysis. Compared to nontreated cells, the expressions of cell adhesion molecules and ECM (extracellular matrix)-receptor interaction molecules were significantly downregulated in EPS364-treated cells (Figure 6A). The expressions of cell adhesion molecules, especially ALCAM (activated leukocyte cell adhesion molecule) and ICAM1 (intercellular adhesion molecule 1), which have been reported to play vital roles in cancer cell adhesion [26,27,28,29], were remarkably downregulated (Figure 6B). Downregulating the expression of adhesion molecules indicated that the cell adhesion capability was reduced. ECM-receptor interaction molecules mediate both the physical linkages with the cytoskeleton and the bidirectional flow of information between the extracellular and intracellular compartments [30]. ECM-receptor interaction molecules have been reported to participate in multiple cancer development pathways [31,32]. Additionally, AFP (α-fetoprotein), which was reported as a biomarker in liver cancer cell growth and apoptosis [33,34], was downregulated after EPS364 treatment. A decreased AFP expression also indicated an antitumor effect of EPS364.

Furthermore, the expression levels of FGFR4 (fibroblast growth factor receptor 4), KLB (β-klotho) and CTNNB1 (β-catenin) were evidently downregulated after the EPS364 treatment (Figure 6B). FGFR4 contains an extracellular domain, a transmembrane domain and an intracellular domain. In the presence of coreceptor KLB, the extracellular domain of FGFR4 binds with another extracellular molecule, FGF19 (fibroblast growth factor 19). FGF19 and its receptor FGFR4 are oncogenic drivers of liver cancer and associated with a poor prognosis [35]. CTNNB1 is generally considered as an intermediate regulator molecule of the FGF19-FGFR4 signaling pathway [28,36,37,38]. In addition, a lot of studies have shown that the downregulation of CTNNB1 was associated with the downregulations of CDH2, ALCAM and ICAM1, which is consistent with our results [31,39,40,41,42,43,44,45,46,47,48]. The extracellular FGF19-FGFR4 signaling pathway mediates the expression of cell adhesion molecules and regulates specific tumorigenic events, including cancer cell proliferation and metastasis, by activating the expression of downstream intracellular genes [35,49,50]. Downregulation of the FGF19-FGFR4 signaling axis induces the generation of ROS and cell apoptosis [51]. As EPS364 downregulated the expression of cell adhesion molecules (Figure 6A,B) and induced cell apoptosis (Figure 5A,B) and the generation of ROS (Figure 5D), we speculated that EPS364 exerted an antitumor activity via targeting the FGF19-FGFR4 signaling pathway.

#### 2.3.4. Quantitative Real-Time PCR (qRT-PCR) and Protein Interaction Analyses

To further elucidate the antitumor mechanism of EPS364, the expression of the genes associated with cancer cell growth, adhesion and the FGF19-FGFR4 signaling pathway were examined using qRT-PCR. These included the FGF19, FGFR4, KLB, AFP, CTNNB1, ALCAM, CDH2 (N-cadherin), CAV1 (caveolin-1), CAV2 (caveolin-2) and ICAM1 encoding genes. The FGF19-FGFR4 signaling pathway has been generally accepted to be an effective and promising therapeutic target in liver cancer treatment [52,53]. Cav-1 and Cav-2 were reported to be regulated by FGFR4 and CDH2 and, also, correlated with tumor progression, invasion and metastasis [54,55,56]. The downregulation of Cav-1 and Cav-2 leads to changes in the cell morphology. As shown in Figure 7A, those genes mentioned above were all downregulated at the transcriptional level after EPS364 treatment in a dose-dependent way.

To verify our speculation that EPS364 inhibits cancer cell growth and adhesion via targeting the FGF19-FGFR4 signaling pathway, the interactions of the cell growth, adhesion and FGF19-FGFR4 signaling pathway-related proteins were detected using the software String 11.0. As shown in Figure 7B, those proteins were closely related, which confirmed our speculations.

Therefore, we proposed a possible antitumor model of EPS364 (Figure 8). In this model, EPS364 downregulated the FGF19-FGFR4 signaling axis. The downregulation of FGF19-FGFR4 signaling induced the collapse of the MMP and the generation of ROS. Besides, the downregulation of the FGF19-FGFR4 signaling pathway was also transferred from CTNNB1 to inhibit the expression of cell adhesion molecule genes such as ICAM1, CDH2 and ALCAM. As a consequence, EPS364 inhibited the cancer cell growth and adhesion.

## 3. Discussion

Liver cancer is a multifactorial disease with multiple biological characteristics, such as cell abnormality, loss of growth control, invasiveness and metastasis [57]. Although several agents against liver cancer are under development, the only effective agents that have been proven to improve survival are sorafenib and regorafenib. Both drugs are oral multi-kinase inhibitors that block RAF (rapidly accelerated fibrosarcoma) signaling, as well as vascular endothelial growth factor, platelet-derived growth factor and KIT. Nevertheless, the pharmacological mechanisms of both drugs are still not well-known, and the prognosis of liver cancer still remains inferior among cancers [2]. Therefore, investigating more anti-liver cancer agents is of great importance in liver cancer treatment. In our previous work, we purified a large quantity of exopolysaccharides from deep-sea extreme environments with antitumor activities. Whether these exopolysaccharides from extreme conditions exhibit similar antitumor mechanisms against liver cancer was not clear. Discovering universal anti-liver cancer mechanisms or targets contributes to new therapeutic strategies and new drug research.

In this work, we purified a novel β-linked polysaccharide EPS364 from deep-sea cold seep conditions. EPS364 possesses a relatively lower Mw of 14.8 kDa, phosphorylation and porous network structure. Notably, our previously studied deep-sea bacterial exopolysaccharide EPS11 possesses an Mw of 22.3 kDa and contains different monosaccharides, including mannose, glucosamine, galacturonic acid, glucose and xylose with a molar ratio of 1:2.58:0.68:0.13:3.09:1.41 [12]. EPS364 and EPS11 possess three identical monosaccharides, including mannose, glucosamine and galacturonic acid, which may contribute to their similar antitumor activities, like inhibiting cancer cell adhesion. Besides, EPS364 exhibited significant antitumor activity. EPS364 attenuated liver cancer Huh7.5 cells filipodia structures and inhibited cell adhesion and growth, similar to that shown in another polysaccharide, EPS11, isolated from extreme deep-sea conditions. Meanwhile, EPS364 induced liver cancer Huh7.5 cell apoptosis, the loss of the MMP and generation of ROS. These results elucidate why EPS364 inhibits cancer cell growth.

To investigate the detailed anticancer molecular mechanisms of EPS364, we performed proteomic and qRT-PCR analyses. Compared with another deep-sea bacterial exopolysaccharide, EPS11, EPS364 also downregulated cancer cell adhesion molecules and ECM-receptor interaction molecules, indicating their similar antitumor styles [11,12]. Of note, unlike EPS11, EPS364 significantly downregulated molecules in the FGF19-FGFR4 signaling pathway, such as FGF19, FGFR4 and KLB. Studies have shown that the FGF19-FGFR4 signaling pathway was an effective antitumor target via mediating a lot of intracellular molecules [52,53]. CTNNB1 as an intermediate molecule between the FGF19-FGF4 signaling pathway and adhesion molecules such as ICAM1, CDH2 and ALCAM [31,39,40,41,42,43,44,45,46,47,48,58] was downregulated after EPS364 treatment. Hence, we speculated that EPS364 might target the FGF19-FGF4 signaling pathway to regulate the expression of adhesion molecules. The interactions of these molecules were verified by protein–interaction analysis String software. Therefore, we concluded that EPS364 inhibits liver cancer cell growth and adhesion via targeting the FGF19-FGF4 signaling pathway.

In summary, a novel exopolysaccharide EPS364 was successfully purified from deep-sea *Vibrio alginolyticus* 364. EPS364 induced cell apoptosis, the loss of the MMP and generation of ROS. Further research revealed that EPS364 inhibited cancer cell growth and adhesion via targeting the FGF19-FGFR4 signaling pathway, which is an effective cancer target [49,51,52,58]. All the aforementioned studies strongly suggest EPS364 as a novel promising antitumor agent for pharmacotherapy. However, to disclose the universal antitumor mechanism of deep-sea exopolysaccharides, more exopolysaccharides need to be purified and studied. In addition, how the deep-sea bacteria synthesize these antitumor exopolysaccharides and secrete extracellularly were not clear. Future research might focus on bacteria exopolysaccharides synthesis and secretion mechanism.

## 4. Materials and Methods

### 4.1. Isolation, Extraction and Purification of Exopolysaccharide EPS364

The bacterial strain *Vibrio alginolyticus* 364 (accession no. MW295527) was isolated from the South China Sea cold seep (119°17′05.3940″ E, 22°06′58.7264″ N) and cultured in 2216E medium (1-g/L yeast extract, 5-g/L tryptone and 1-L freshly filtered seawater, pH 7.4–7.6) at 28 °C for 4 days.

The crude polysaccharide was obtained by ethanol precipitation [15]. The culture supernatant was concentrated by centrifugation at a speed of 10,000 rpm at 20 °C for 20 min, and the polysaccharide was precipitated with four volumes of 95% (*v*/*v*) ethanol. After keeping at 4 °C overnight, the precipitate was collected by centrifugation with a speed of 10,000 rpm at 4 °C for 20 min and dissolved in distilled water. To obtain high-purity polysaccharide, the crude polysaccharide was further successively separated by a DEAE Sepharose Fast Flow column (Solarbio^®^ Life Sciences, Beijing, China) and then Sephadex G-100 column (Solarbio^®^ Life Sciences, Beijing, China). After lyophilization of the final active fractions, pure EPS364 powder was obtained and then weighed for further research. All fractions were detected using the method of a phenol-sulfuric acid assay [15]. The growth inhibition activity against Huh7.5 cancer cells was detected by the MTT (3-(4,5)-dimethylthiahiazo (-z-y1)-3,5-di-phenytetrazoliumromide) assay, as described below. The purity of EPS364 (1.0 mg/mL) was detected using a NanoPhotometer^®^ spectrophotometer (IMPLEN, Westlake Village, CA, USA) at a range of 200–800 nm.

### 4.2. Characterization of Polysaccharide EPS364

#### 4.2.1. Molecular Weight Analysis

The molecular weight of exopolysaccharide EPS364 was examined by high-performance gel permeation chromatography (HPGPC) with a Shodex SUGAR KS-804 column (300 mm × 8.0 mm, 7 μm) (Shodex, Tokyo, Japan) [59]. Dextrans (9750 Da, 13,050 Da, 36,800 Da, 64,650 Da, 135,350 Da and 300,600 Da) were used as the standards.

#### 4.2.2. Monosaccharide Composition Analysis

EPS364 powder was dissolved in trifluoroacetic acid (2.0 mol/L) at a concentration of 5.0 mg/mL, followed by heating at 110 °C for 4 h to release the component monosaccharides. The excessive acid was neutralized with a 4-mol/L NaOH solution. The monosaccharide composition analysis of EPS364 was performed using PMP (1-phenyl-3-methyl-5-pyrazolone) as a precolumn derivatization reagent [60]. The Agilent 1260 HPLC system (Agilent, Santa Clara, CA, USA) equipped with a ZORBAX SB-AQ C18 column (4.6 mm × 250 mm, 5 µm) was used to analyze the monosaccharide composition of EPS364. Mannose, glucosamine, rhamnose, gluconic acid, galacturonic acid, galactosamine, glucose, galactose, xylose and arabinose were used as the standards. The monosaccharide composition of EPS364 was determined by comparison with the retention time of standard monosaccharides.

#### 4.2.3. Fourier-Transformation Infrared (FTIR) Spectroscopy Analysis

EPS364 powder was pressed into pellets and then detected using a Fourier-transform infrared spectrophotometer (Nicolet iS50, Thermo, Waltham, MA, USA) ranging from 4000 to 500 cm^−1^.

#### 4.2.4. Scanning Electron Microscopy (SEM) and Energy-Dispersive Spectrum (EDS) Analyses of Polysaccharide EPS364

EPS364 power was sputter-coated with gold and platinum using an Ion Sputter (MC1000, Hitachi, Tokyo, Japan). For the SEM observations, the EPS364 specimen was observed using a scanning electron microscope (Hitachi S-3400N, Japan) at an accelerating voltage of 5 kV. For the EDS analysis, the EPS364 specimen was detected using an energy-dispersive spectrum (model 550i, IXRF Systems, Austin, TX, USA) at an accelerating voltage of 5 keV for 30 s [61].

### 4.3. Measurement of Anticancer Activities

#### 4.3.1. Cell Culture

The liver cancer Huh7.5 cell line, human normal liver HL-7702 cell line and liver cancer Bel-7402 cell line were purchased from the American Type Culture Collection (Manassas, VA, USA). The Huh7.5 Cells and Bel-7402 cells were cultured in RPMI (Roswell Park Memorial Institute)-1640 medium containing 10% FBS (fetal bovine serum), 100-µg/mL streptomycin and 100-U/mL penicillin at 37 °C with 5% CO_2_. HL-7702 cells were cultured in DMEM (Dulbecco’s Modified Eagle Medium) medium containing 10% FBS, 100-µg/mL streptomycin and 100-U/mL penicillin at 37 °C with 5% CO_2_. 

#### 4.3.2. Cell Proliferation Assay

The growth of Huh7.5 liver cancer cells was evaluated using the MTT method [62]. Huh7.5 cells were treated with different concentrations of EPS364 for 48 h. Except for the treatment time effect test, the cells were separately treated with 0.5-mg/mL EPS364 for 24, 48 and 72 h. After incubation with MTT (5 mg/mL, 20 μL/well) for another 4 h, 100-µL “Triplex Solution” (10% SDS-5% isobutanol and 12-mM HCl) was added to each well. Finally, the absorbance of each well was measured at 570 nm by a multi-detection microplate reader (Infinite M1000 Pro, TECAN, Mannedorf, Switzerland). The inhibition rate was presented as a percentage relative to the control group. All experiments were performed three times.

#### 4.3.3. Cell Morphology Observation by Inverted Phase Contrast Microscope and Scanning Electron Microscopy (SEM)

To observe the cell morphology changes after EPS364 treatment, liver cancer Huh7.5 cells and the human normal liver HL-7702 cell line were treated with different concentrations of EPS364 for 12 h. The cell morphology was observed by an inverted-phase contrast microscope (NIKON TS100, Tokyo, Japan) and scanning electron microscopy using our previous method [11]. For the inverted-phase contrast microscope observations, cells on plates were observed by an inverted-phase contrast microscope. For the SEM observations, the cells on glass coverslips were visualized using SEM (Hitachi S-3400N, Tokyo, Japan).

#### 4.3.4. Cell Apoptosis, Mitochondrial Membrane Potential (MMP) and Reactive Oxygen Species (ROS) Detection

Huh7.5 cells (1.5 × 10^5^ cells) were seeded into a six-well plate and then treated with different concentrations of EPS364 (0–2 mg/mL) for 12 h. For apoptosis detection, cells were incubated with Annexin V-FITC (Fluorescein Isothiocyanate) and PI (propidium iodide, Nanjing KeyGEN Biotech Co., Ltd., Jiangsu, China). For the MMP detection, cells were incubated with JC-1 (Solarbio^®^ Life Sciences, Beijing, China). For ROS detection, cells were stained with DCFH-DA (2,7-Dichlorodi-hydrofluorescein diacetate, Beyotime, Shanghai, China). Finally, cells were analyzed by flow cytometry (FACS Aria™ II, BD, San Jose, CA, USA) [63]. All experiments were performed three times.

#### 4.3.5. Proteomic Analysis

A proteomic analysis was performed by PTM Biolabs Co., Ltd. (Hangzhou, China) [11]. Briefly, Huh7.5 cells were treated with or without 4.0-mg/mL EPS364 for 12 h, respectively. Cells were collected by centrifugation followed by lysis to obtain the total cellular proteins. Protein samples were labeled and then quantified by a LC-ESI-MS/MS analysis. All MS data were deposited to the ProteomeXchange Consortium via the PRIDE partner repository with the dataset identifier PXD023080. Finally, the protein ratios of the treated group/control group in different functional classifications were analyzed.

#### 4.3.6. Quantitative Real-Time PCR (qRT-PCR) and Protein Interaction Analyses

Huh7.5 cells were treated with different concentrations of EPS364 (0, 0.8 or 1.2 mg/mL) for 5 h. The total RNA of the Huh7.5 cells was extracted using a Trelief™ RNAprep FastPure Tissue&Cell kit (Tsingke Biotechnology Co., Ltd., Beijing, China). RNA was reversely transcribed into cDNA using a ReverTra Ace qPCR RT Master Mix with a gDNA Remover kit (Toyobo Co., Ltd., Osaka, Japan). The PCR amplification was carried out using a SYBR^®^ Green Realtime PCR Master Mix (Toyobo Co., Ltd., Osaka, Japan), and the transcriptional levels of different genes were detected by an ABI 7900 real-time PCR system (Applied Biosystems, Foster City, CA, USA). The relative expression was calculated using the 2^−∆∆Ct^ method. All primers are listed in Appendix A. All experiments were performed three times.

The interactions of the proteins were analyzed using the online software String 11.0 (https://string-db.org/ (accessed on 17 December 2020)).

### 4.4. Statistical Analysis

All data were presented as the means ± standard error. The statistical analysis was performed using IBM SPSS Statistics 26 (IBM Corporation, Armonk, NY, USA) software. *p*-values less than 0.05 were considered statistically significant (* *p* < 0.05 and ** *p* < 0.01).

## Figures and Tables

**Figure 1 marinedrugs-19-00171-f001:**
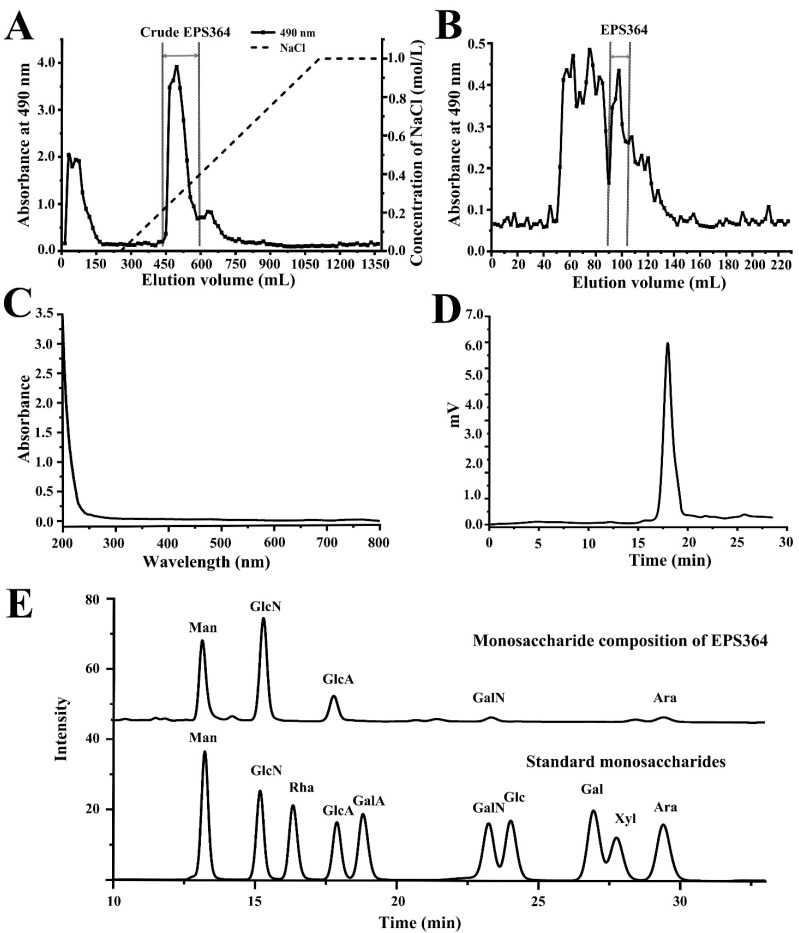
Purification and composition identification of EPS364. (**A**) The chromatograph profile of the crude polysaccharide EPS364 on a DEAE Sepharose Fast Flow column eluted with a linear gradient of NaCl aqueous solutions (0–1.0 M) at a flow rate of 2.0 mL/min. (**B**) The elution profile of EPS364 on a Sephadex G-100 column eluted with ultrapure water at a flow rate of 1.0 mL/min. (**C**) UV spectrum of EPS364 detected by a NanoPhotometer^®^ spectrophotometer. (**D**) HPGPC (high-performance gel permeation chromatography) profile of EPS364. (**E**) The monosaccharide composition of EPS364. Abbreviations: Man, mannose; GlcN, glucosamine; Rha, rhamnose; GlcA, gluconic acid; GalA, galacturonic acid; GalN, galactosamine; Glc, glucose; Gal, galactose; Xyl, xylose and Ara, arabinose.

**Figure 2 marinedrugs-19-00171-f002:**
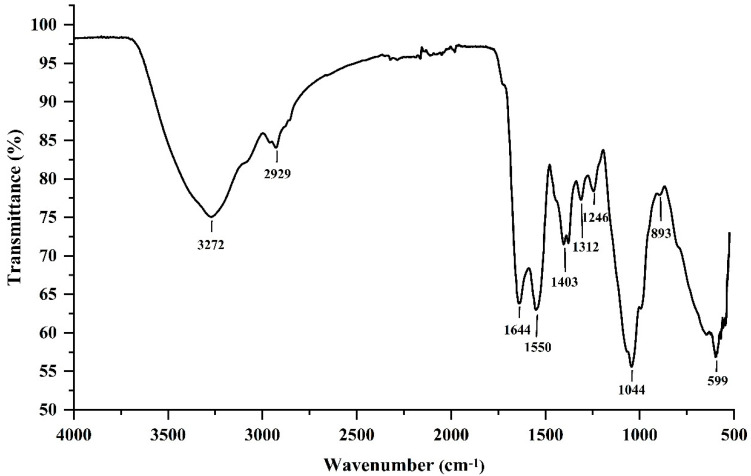
Fourier-transform infrared (FTIR) spectrum of polysaccharide EPS364.

**Figure 3 marinedrugs-19-00171-f003:**
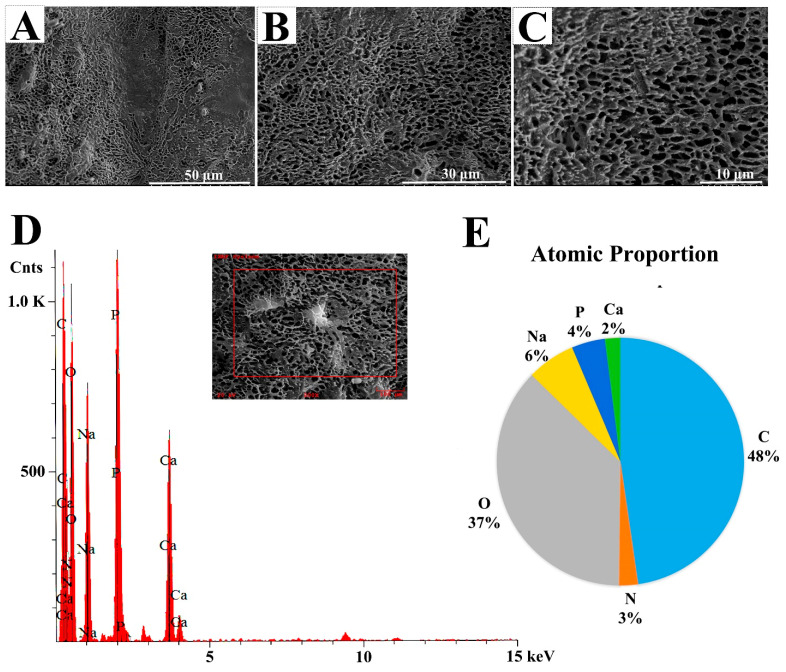
SEM analysis of EPS364 with amplification of 1000× (**A**), 1700× (**B**) and 3000× (**C**). (**D**) Energy-dispersive spectrum (EDS) analysis of EPS364. (**E**) Atomic proportion of EPS364 detected by EDS.

**Figure 4 marinedrugs-19-00171-f004:**
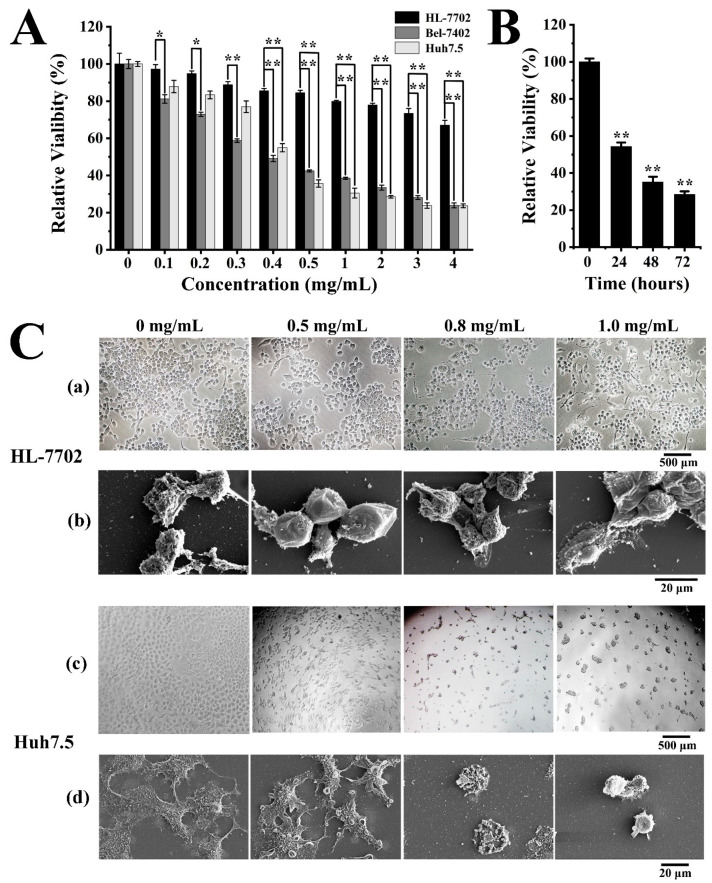
EPS364 inhibited Huh7.5 cells proliferation and adhesion. (**A**) Cell viability after different concentrations of the EPS364 (0–4 mg/mL) treatment for 48 h detected by MTT (3-(4,5)-dimethylthiahiazo (-z-y1)-3,5-di-phenytetrazoliumromide) assay. (**B**) Huh7.5 cell viability after EPS364 treatment (0.5 mg/mL) for different times (0–72 h) detected by the MTT method. (**C**) The cell morphology after EPS364 treatment for 12 h. (**a**) HL-7702 cell morphology observed by inverted-phase contrast microscope, (**b**) HL-7702 cell morphology observed by scanning electron microscope (SEM), (**c**) Huh7.5 cell morphology observed by inverted-phase contrast microscope and (**d**) Huh7.5 cell morphology observed by scanning electron microscope (SEM). The cell viability assays were performed in three dependent experiments. * *p* < 0.05 and ** *p* < 0.01 versus control.

**Figure 5 marinedrugs-19-00171-f005:**
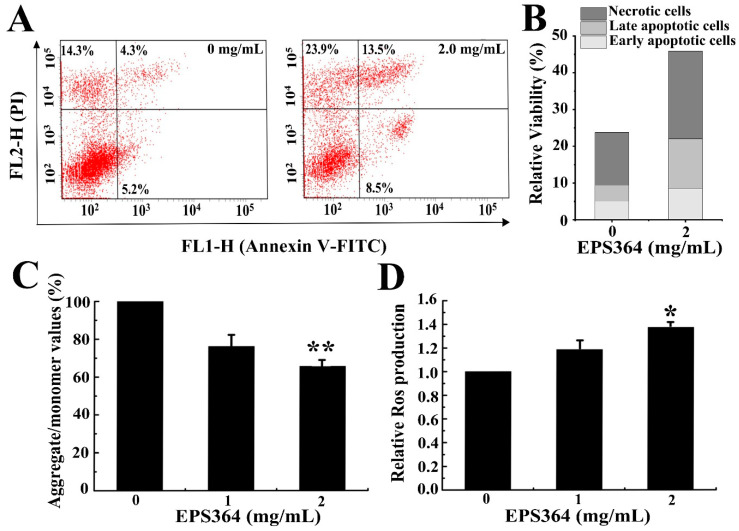
EPS364 induced apoptosis, the loss of the mitochondrial membrane potential (MMP) and generation of reactive oxygen species (ROS) in Huh7.5 cells. After incubation with or without the EPS364 treatment (2 mg/mL) for 12 h, Huh7.5 cells were stained by Annexin V-FITC (fluorescein isothiocyanate) and PI (propidium iodide) and detected by a flow cytometry. (**A**) Representative dot plots of Annexin V/PI staining. (**B**) Column bar graph of apoptotic cells. (**C**) EPS364 induced the loss of the MMP in Huh7.5 cells with 12 h of treatment. (**D**) EPS364 induced the generation of ROS in Huh7.5 cells with 12 h of treatment. * *p* < 0.05 and ** *p* < 0.01 versus the control.

**Figure 6 marinedrugs-19-00171-f006:**
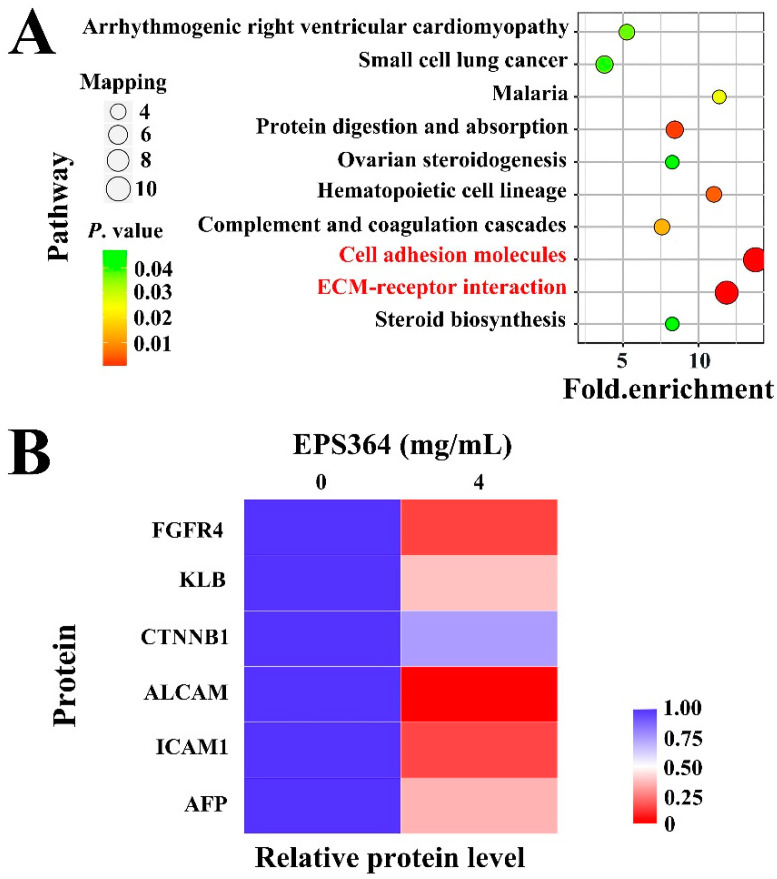
Proteomic analysis of Huh7.5 cells treated with EPS364 for 12 h. (**A**) KEGG (Kyoto Encyclopedia of Genes and Genomes) enrichment of the downregulated pathways in EPS364-treated cells. The cell adhesion molecules and extracellular matrix (ECM)-receptor interaction molecules were significantly downregulated in EPS364-treated cells. (**B**) Heatmap showing the representative proteins related to cancer cell growth, adhesion and the FGF19-FGFR4 signaling pathway that were downregulated in the EPS364-treated group. Abbreviations: FGFR4, fibroblast growth factor receptor 4; KLB, β-klotho; CTNNB1, β-catenin; ALCAM, activated leukocyte cell adhesion molecule; ICAM1, intercellular adhesion molecule 1 and AFP, α-fetoprotein.

**Figure 7 marinedrugs-19-00171-f007:**
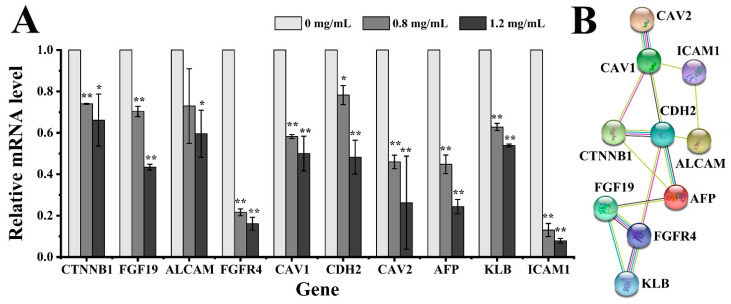
Expression and interaction assays of proteins that were significantly downregulated by the treatment of EPS364 for 5 h, as shown in the proteomic results. (**A**) The relative mRNA levels of different genes related to cancer cell growth, adhesion and the FGF19-FGFR4 signaling pathway were detected by qRT-PCR after different concentrations of the EPS364 (0–1.2 mg/mL) treatment. (**B**) The interactions of the cancer cell growth, adhesion and FGF19-FGFR4 signaling pathway-related proteins were analyzed by String software. Data are presented as means ± SD of three independent experiments (*n* = 3). * *p* < 0.05 and ** *p* < 0.01 versus the EPS364-treated group. Abbreviations: FGFR4, fibroblast growth factor receptor 4; KLB, β-klotho; CTNNB1, β-catenin; ALCAM, activated leukocyte cell adhesion molecule; ICAM1, intercellular adhesion molecule 1; AFP, α-fetoprotein; CAV1, caveolin-1; CAV2, caveolin-2; CDH2, N-cadherin and FGF19, fibroblast growth factor 19.

**Figure 8 marinedrugs-19-00171-f008:**
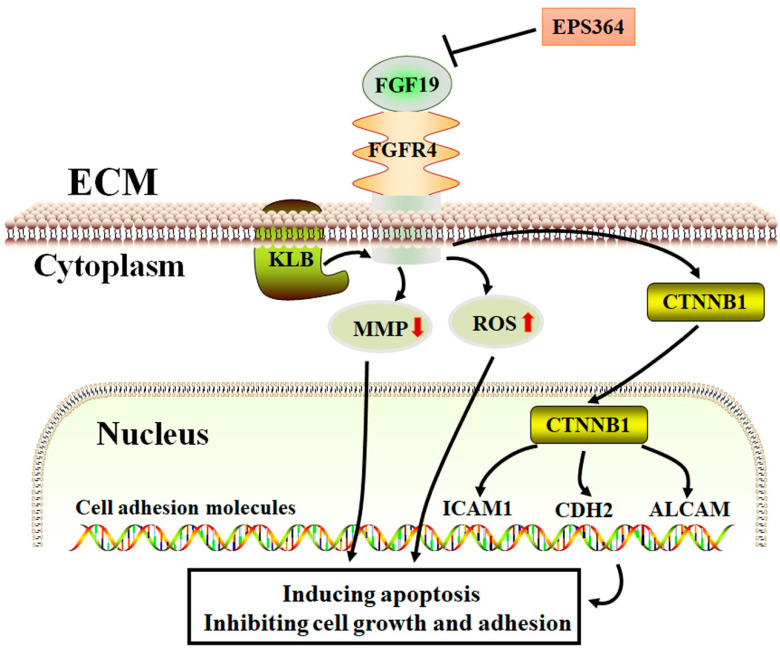
Proposed antitumor model of EPS364. EPS364 downregulated the FGF19-FGFR4 signaling axis, which induced the collapse of the MMP and generation of ROS. FGF19-FGFR4 signaling downregulated the downstream molecules, such as CTNNB1, ICAM1, CDH2 and ALCAM. Consequently, EPS364 induced apoptosis and inhibited cell growth and adhesion.

## Data Availability

The data is contained within the article or the Appendix A.

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
