# Peer review of "EPS364, a Novel Deep-Sea Bacterial Exopolysaccharide, Inhibits Liver Cancer Cell Growth and Adhesion"

_marinedrugs, 2021, doi:10.3390/md19030171_

Round 1

Reviewer 1 Report

This manuscript by Wang et al details the purification, characterisation and assay of an exopolysaccharide derived from deep sea bacteria. EPS364 is active against a liver cancer cell line and was shown to induce apoptosis, affect mitochondrial membrane potential and generate reactive oxygen species in these cells. Proteomic and genomic analyses suggest involvement of the FGF19-FGFR4 signaling pathway in the mechanism of action of EPS364.

The characterisation and physical properties of the polysaccharide has been examined by several different methods yet its primary structure is not determined. Surely this is important to demonstrate its potential as an anti-tumor agent. In addition, it is disappointing that EPS364 is only tested upon a single liver cancer cell line, without extension to other cancer cell types or indeed heathy non-cancerous cells. A claim of ‘anti-cancer’ activity should not be made without evidence of selectivity.

In general, the manuscript is difficult to read with poorly defined acronyms and poorly labelled figures. It would greatly benefit from professional English writing assistance to correct the many instances of poor grammar and language.

Specific comments

Title: The title should reflect the focus of this work on liver cancer rather than generalising to all cancers.

Introduction: Several phrases are too vague and should be clarified to strengthen the introduction eg. ‘and a lot’ line 40 ; ‘a variety of underlying pharmaceutical activities’ line 43. The precise focus and aims of this project are unclear.

Results: Comparison to previous studies does not belong in the results section eg. Lines 83-88 should be moved to the discussion section. Similarly for lines 208-216. Results should be limited to presentation of data without summarising relevance or discussion.

Figure 1: please define HPSEC. Please elaborate on the figure legend for (E). what do the two traces represent? What is the intensity unit? What are the impurities seen in the top trace? Please define the monosaccharide abbreviations in the figure legend.

2.2.3 please define SEM and EDS. What ‘characteristics of a polymer’ are being referred to in line 119?

Figure 3: the labels on Figure 3D cannot be read. What is the square box of 3D? perhaps D could be made bigger and E could be reduced in size.

The paragraph following Figure 3 is unnecessary here. The meaning of lines 141-2 is unclear. Lines 144-6 are a repetition of lines 121-3. The claims of water solubility, viscosity and water holding capacity of the polysaccharide are ambitious without experimental evidence of these properties.

Figure 4: how many replicates were tested? Please define the asterisks in the legend. What are they relevant to? Is it simply each concentration relative to 0mg/mL without comparing individual concentrations?

2.3.2 please define MMP. What is meant by ‘primal changes of MMP and ROS’ line 177?

The text (lines 186 and 192) claim a dose dependent effect of EPS364 on MMP and ROS but the data only shows significance at 2mg/mL. Dose dependence is not proven by this data.

Different dosage treatments for different tests is confusing. For example, Figure 4A shows minimal difference between 0.5 – 4 mg/mL yet Fig 5A and B use 2mg/mL, Fig 5 C and D use 1 and 2 mg/mL and figure 6 uses 4 mg/mL. Figure 7 changes further to 0.8 and 1.2 mg/mL. Please clarify these choices.

Figure 5. how many replicates were tested? Why does Fig 5B show necrotic cells at 0mg/mL yet Fig4A shows 100% viability ?

Figure 6. please define the proteins listed in Fig 6B. what are the units of the colour coding?

Figure 7. why were these dosages chosen? There appears to be no significant differences between them. How many replicates were tested? Why was the time only 5 hrs whereas other tests were done for at least 12 hrs? please define the proteins listed.

Methods: The heading of 4.2.2 is the same as 4.2.1. There is no mention of replicates. The control group appears to be treatment of the Huh7.5 cells with no polysaccharide. The effect of polysaccharide should be tested on healthy cells at a minimum (and extended to other cancer cell lines) to show selectivity.

Reviewer 2 Report

I write you in regards to manuscript entitled “EPS364, a Novel Deep-Sea Bacterial Exopolysaccharide, Inhibits Cancer Cell Growth and Adhesion” which you submitted to Marine drugs.

As author notes in this report, this study might provide useful information on a novel exopolysaccharide for pharmacotherapy. The paper provides interesting data but it still needs a considerable revision to be acceptable for Marine drugs.

Major comments

・In vitro studies, the cytotoxicity has been observed with the high concentration of EPS364. Is osmotic pressure associated with cell death? Also, how do you deliver the high concentrations of ESP364 to liver cancer?

・In figure 5, the rate of apoptotic cells and ROS positive cells is not a high and the rate of necrotic cells is high. I think that mechanism of cell death induced by EPS364 might be necrosis.

Reviewer 3 Report

The manuscript entitled “EPS364, a Novel Deep-Sea Bacterial Exopolysaccharide, Inhib- 2

 its Cancer Cell Growth and Adhesion” written by Wang and co-workers describes on purification of a new exopolysaccharide named EPS365 suggesting its structural property and anticancer activity. Detailed anti-liver-cancer evaluation and adhesion was described on the purified EPS365. Information involved in this manuscript is important in new anticancer drug discovery. Therefore, the reviewer suggests this manuscript to be published in marine drugs after minor revision. Please revise along points listed below.

  1. Please provide full spelling when abbreviation was appeared at the first time, EDS in page 4 line 124, HPSEC in page3 Fig. 1D, SEM, EDS in Page 4 L120, 124, and HPGPC in page 11 line 343, for examples.
  2. Page 12 L1, title of this section is not “molecular weight analysis.”
  3. Page 2 lines 78-83; it not good sentences for understanding. Suddenly, different content “we concluded ……” comes. Describe how monosaccharide composition was determined.
  4. Page 3 Figure 1D; the peak corresponding EPS364 looks to have a shoulder as well as Figure 1A; 1D : mV; note what two chromatograms in Figure1E are expressing.
  5. Page 4 l118 has two periods after section number.
  6. P11; Experimental; isolation section doesn’t include information for weight along isolation.

Reviewer 4 Report

The presented paper is an original and innovative research. The well-described methodology is unqualified. However, the authors of the work disturbed the proper structure of the research work: lack of a precisely described goal in the introduction; the research methodology should be presented successively, then the results and finally a discussion. I am asking for a proper structure of the research work. A review by a native speaker is recommended.

Round 2

Reviewer 1 Report

This revised manuscript by Wang et al is greatly improved by the additional experiments, and editing of figures and text. I have no further concerns and recommend it for publication in Marine Drugs.

Reviewer 2 Report

The manuscript has been revised well. I think this manuscript is acceptable.